# High-Dimensional Cytometry Dissects Immunological Fingerprints of Idiopathic Inflammatory Myopathies

**DOI:** 10.3390/cells11203330

**Published:** 2022-10-21

**Authors:** Christopher Nelke, Marc Pawlitzki, Christina B. Schroeter, Niklas Huntemann, Saskia Räuber, Vera Dobelmann, Corinna Preusse, Andreas Roos, Yves Allenbach, Olivier Benveniste, Heinz Wiendl, Ingrid E. Lundberg, Werner Stenzel, Sven G. Meuth, Tobias Ruck

**Affiliations:** 1Department of Neurology, Medical Faculty, Heinrich Heine University Dusseldorf, 40225 Dusseldorf, Germany; 2Department of Neurology with Institute of Translational Neurology, University Hospital Münster, 48149 Münster, Germany; 3Department of Neuropathology, Charité-Universitätsmedizin Berlin, Corporate Member of Freie Universität Berlin and Humboldt-Universität zu Berlin, 10117 Berlin, Germany; 4Department of Neuropediatrics, University of Duisburg-Essen, 45147 Essen, Germany; 5Service de Médecine Interne et Immunologie Clinique, University Hospital Pitié Salpêtrière, 75013 Paris, France; 6Division of Rheumatology, Department of Medicine, Solna, Karolinska Institutet, and Karolinska University Hospital, 171 77 Stockholm, Sweden

**Keywords:** inflammatory idiopathic myopathy, myositis, immune signature, cytometry, flow cytometry

## Abstract

Chronic inflammation of skeletal muscle is the common feature of idiopathic inflammatory myopathies (IIM). Given the rarity of the disease and potential difficulty of routinely obtaining target tissue, i.e., standardized skeletal muscle, our understanding of immune signatures of the IIM spectrum remains incomplete. Further insight into the immune topography of IIM is needed to determine specific treatment targets according to clinical and immunological phenotypes. Thus, we used high-dimensional flow cytometry to investigate the immune phenotypes of anti-synthetase syndrome (ASyS), dermatomyositis (DM) and inclusion-body myositis (IBM) patients as representative entities of the IIM spectrum and compared them to healthy controls. We studied the CD8, CD4 and B cell compartments in the blood aiming to provide a contemporary overview of the immune topography of the IIM spectrum. ASyS was characterized by altered CD4 composition and expanded T follicular helper cells supporting B cell-mediated autoimmunity. For DM, unsupervised clustering identified expansion of distinct B cell subtypes highly expressing immunoglobulin G4 (IgG4) and CD38. Lastly, terminally differentiated, cytotoxic CD8 T cells distinguish IBM from other IIM. Interestingly, these terminally differentiated CD8 T cells highly expressed the integrin CD18 mediating cellular adhesion and infiltration. The distinct immune cell topography of IIM might provide the framework for targeted treatment approaches potentially improving therapeutic outcomes.

## 1. Introduction

The spectrum of idiopathic inflammatory myopathies (IIM) is characterized by chronic inflammation of skeletal muscle, resulting in progressive muscle weakness as clinical hallmark [1,2]. On the basis of clinical presentation, histopathological features and antibody status, IIM have been classified in distinct, but often overlapping, disease entities, e.g., polymyositis (PM), dermatomyositis (DM) and inclusion-body myositis (IBM) [1].

The burden of disease is amplified by extra-muscular affection commonly present in IIM [3]. Consequently, IIM has evolved into a systemic inflammatory disease spectrum involving multiple organs. Among these, IBM is unique due to its characteristic clinical presentation, treatment refractory disease course and controversial viewpoints regarding its pathogenesis [4]. Exemplifying IIM with predominance of extra-muscular features, anti-synthetase syndrome (ASyS) is distinguished by myositis, interstitial lung disease, mechanic’s hand and arthritis [3,5]. While our understanding of immunological hallmarks of certain IIM has evolved over the past decade, the immune signatures driving chronicity of IIM remain largely enigmatic. Consequently, treatment approaches are non-specific often combining glucocorticoids with broad immunosuppressive agents [6]. To utilize contemporary therapeutics targeting specific disease pathways, a deeper understanding of drivers of disease is required for IIM.

Therefore, we chose a high-dimensional flow cytometry approach to investigate immune phenotypes of IIM. The strategy of high-dimensional flow cytometry has been previously employed to successfully dissect immunotypes of inflammatory diseases such as coronavirus disease 2019 [7]. To further harness the value of this approach, we decided to include three cohorts of patients, representative of the IIM spectrum. We included DM as characteristic, antibody-mediated IIM with a focus on skeletal muscle and skin, ASyS as overlap syndrome with muscular and common extra-muscular affection and IBM with predominance/exclusivity of skeletal muscle affection. Our analysis revealed a distinct immunological topography of IIM entities and might instruct specific treatment strategies.

## 2. Material and Methods

### 2.1. Patients and Data Collection

Ninety patients with IIM (ASyS: *n* = 41, DM: *n* = 28; IBM: *n* = 21), were included in our study between January 2017 and December 2020. Inclusion required established diagnosis of IIM according to the European League Against Rheumatism (EULAR) and the American College of Rheumatology (ACR) Classification Criteria [6] and with respect to the recommendations of the 239th European Neuromuscular Centre (ENMC) international workshop [8]. At inclusion, the existing immunotherapies were as low as possible and prescribed at a stable dose for at least 3 months for all patients. Twelve healthy controls were included in this study. Inclusion required that subjects had no known diseases and normal creatine kinase (CK) level at the time of blood sampling. Peripheral blood was collected from all participants after obtaining informed consent. Clinical and demographic data were acquired from the electronic medical records and standardized into case report forms for down-stream analysis. The local ethics committee (2016-053-f-S and 2021-1417) approved the study. This trial was conducted in accordance with the Declaration of Helsinki.

### 2.2. Sample Processing and Antibody Staining

Whole blood samples were collected, and peripheral mononuclear cells (PBMC) were isolated by Ficoll (Sigma-Aldrich, St. Louis, MO, USA) density gradient centrifugation and stored in liquid nitrogen according to our standard operating procedure (SOP) until usage [9]. For analysis, freshly thawed PBMCs were centrifuged at 300× *g* for 5 min. PBMCs were then resuspended in phosphate buffered saline (PBS, Sigma-Aldrich) supplemented with 2% heat-inactivated foetal bovine serum (FBS, GE Healthcare, Chicago, IL, USA) and 2 mM ethylenediaminetetraacetic acid (EDTA, Sigma-Aldrich). For antibody staining, PBMCs were then incubated with fluorochrome-conjugated antibodies at 4 °C for 30 min in the dark (Appendix A). For intracellular staining, cells were treated with fixation/permeabilization solution (Thermo Fisher Scientific, Waltham, MA, USA) for 20 min, washed with permeabilization buffer (Thermo Fisher Scientific) and incubated with antibodies directed against intracellular target molecules of interest. For cytokine analysis, PBMCs were cultivated overnight in X-VIVO™ 15 Serum-free Hematopoietic Cell Medium (Lonza, Basel, Switzerland). Next, PBMC were stimulated with Leucocyte Activation Cocktail, with BD GolgiPlug™ with added phorbol myristate acetate, ionomycin and brefeldin A (BD Biosciences, Heidelberg, Germany) for 4 h before staining. PBMC were stained with extracellular lineage markers for identification and intracellular staining for the corresponding cytokine. Approximately 1 × 10^6^ to 3 × 10^6^ freshly isolated PBMCs were used per sample and staining. When indicated, PBMCs were washed and additionally stained with the Zombie NIR or Aqua Fixable Viability Kit (Biolegend, San Diego, CA, USA) according to the manufacturer’s manual to distinguish between dead and live cells. PBMCs were washed and resuspended in PBS/FBS/EDTA and analyzed by flow cytometry using a CytoFlex Flow Cytometer (Beckman Coulter, Brea, CA, USA).

### 2.3. High-Dimensional Data Analysis of Flow Cytometry Data

viSNE and FlowSOM analyses were performed with Omiq^®^ (Dotmatics, Boston, MA, USA). B cells, CD4^+^ T cells, and CD8^+^ T cells were analysed separately. For viSNE analysis, 15,000 cells were randomly sampled from each FCS file with a theta of 0.5, a perplexity of 30 and 5000 iterations. Optimization of tSNE (opt-sne) algorithm was used for dimensional reduction as previously described [10]. For visual exploration, opt-sne displays each cell two-dimensional graphs by maximizing the probability that similar cells are grouped together. Similarity is computed by comparing the pattern of markers measured by flow cytometry. On the right side of each graph, a z-scale displays the relative level of the flow cytometry marker indicated above each graph [10].

For unbiased clustering, we used the established FlowSOM algorithm [11]. Briefly, FlowSOM is centred around a self-organizing map (SOM) consisting of nodes in a multidimensional space. The SOM is trained on random datapoints (cells). Here, similar datapoints are grouped to their nearest neighbour with each consecutive datapoint extending the training SOM. In traditional FlowSOM analysis, a minimum spanning tree is used to visualize the SOM. The algorithm restricts the maximum number of neighbouring nodes resulting in a tree in which nodes are connected with similar datapoints. Finally, SOM clusters are aggregated into meta-clusters based on consensus hierarchical clustering. Bidimensional scatter plots are then inspected to validate the results. For FlowSOM analysis of the B cell compartment, the following markers were used for clustering: IgM, IgD, IgG1, IgG4, CD21, CD24, CD27, CD38 and CD80. A total of 115 clusters were obtained. Consensus hierarchical clustering yielded 12 meta-clusters. Next, we entered the aforementioned B cell markers into an opt-sne algorithm to draw a two-dimensional graph of the dataset. Each datapoint (cell) was labeled according to the corresponding meta-cluster.

For PCA, each patient was treated as one data point. Data were scaled and centred.

We used biparametric gating for quantification of established immune cell populations. The use of biparametric gating allows for comparison and replication of data across different studies and research groups. For visual inspection, we displayed immune cell compartments as opt-sne graph. This approach allows for dimensionality reduction and for visualizing marker expression for specific immune cell populations. Finally, we applied FlowSOM as an unbiased clustering algorithm to the B cell compartment. The use of FlowSOM is motivated by a lack of consistent classification of B cell populations and subpopulations according to flow cytometry markers [12]. By using an unbiased clustering algorithm, we aimed to distinguish individual B cell populations based on a distinct set of multiparametric surface markers. An overview of all flow cytometry panels used in this manuscript is given in Appendix A.

### 2.4. Statistical Analysis

Statistical Analysis was performed using GraphPad Prism 9.2 (GraphPad Software, Inc., San Diego, CA, USA) and R 3.5.3 (R Foundation for Statistical Computing, Vienna, Austria). Given the explorative approach and data heterogeneity, nonparametric tests for detecting group differences were used throughout unless indicated otherwise. Group differences were assessed by unpaired Wilcoxon test (for 2 categories) or Kruskal–Wallis test (for more than 2 categories). For linear regression, Goodness-of-fit was assessed by Tjur’s R squared, significance by log-likelihood test. Differences were considered statistically significant with the following *p*-values: * *p* < 0.05, ** *p* < 0.01, *** *p* < 0.001 and **** *p* < 0.0001.

### 2.5. Data Availability

All data associated with this study are present in the paper. The data assessed in this study are available from the corresponding author on reasonable request.

## 3. Results

### 3.1. Clinical Spectrum of IIM

The demographics, clinical and laboratory data of the study cohort are provided in Table 1. It should be noted that important clinical characteristics differed between the cohorts (Figure 1A). These included patient age, as the IBM cohort was older than DM, ASyS and healthy controls. Regarding muscle function, IBM patients were more severely affected as evidenced by Manual Muscle Testing (MMT)-8 examination as compared to ASyS patients. A subgroup of patients in all three groups did not receive any immunotherapy at time of blood sampling, while the remaining patients received immunosuppressive therapies (Table 1). We selected these cohorts as they reflect clinical and epidemiological characteristics previously described for the IIM spectrum [3,13,14].

### 3.2. Broad Immune Cell Populations Are Unchanged in IIM Entities

To begin investigation of circulating immune cell populations of IIM, we analysed PBMCs of ASyS, DM and IBM patients as well as healthy controls using high-dimensional flow cytometry (Figure 1B). First, we investigated a panel focused on major lymphocyte populations. Life PBMC were gated for lymphocytes and cell numbers of B cells and T cell subsets were recorded (Figure 2A). Here, we observed no meaningful differences for broad immune cell subsets including B cells, T cells and CD4^+^ and CD8^+^ T cells among different IIM entities or compared to healthy control (Figure 2B). Aiming to understand the immune topography, we applied high-dimensional mapping of the flow cytometry data using opt-sne. For dimensional reduction, 15,000 cells were randomly sampled from each patient. We included standard lymphocyte markers for B and T cells (CD3, CD4, CD8, CD19 and CD20). Cells were then concatenated and overlaid. Key regions were highlighted by CD19^+^, CD4^+^ and CD8^+^ expression levels for individual cells corresponding to the respective immune cell population (Figure 3C). This approach allowed us to visualize approximately 1,350,000 cells in a two-dimensional space and to compare datasets of IIM subgroups. It is interesting to note that immune cell subsets were largely similar among IIMs when using standard lymphocyte markers with only subtle differences observed among the entities. As broad immune cell patterns appear unsuitable to differentiate ASyS, DM, IBM or healthy controls, we decided to investigate individual lymphocyte compartments.

### 3.3. IBM Is Characterized by a Terminal CD8 T Cell-Pathology

First, we focused on the CD8 T cell compartment. We investigated quantitative and qualitative changes to four major CD8 T cell subsets defined as naïve (CD3^+^, CD8^+^, CD27^+^, CD45RA^+^/CD45RO^−^), central memory (CD3^+^, CD8^+^, CD27^+^, CD45RA^−^/CD45RO^+^ [CM]), effector memory (CD3^+^, CD8^+^, CD27^−^, CD45RO^+^ [EM]) and effector memory T cells re-expressing CD45RA (CD3^+^, CD8^+^, CD27^−^, CD45RA^+^/CD45RO^−^ [TEMRA]) (Figure 3A). We focused our analysis on the relative immune cell distribution. Naïve, CM and EM CD8 T cell populations remained unchanged, while TEMRA cells were substantially expanded in the blood of IBM patients as compared to ASyS, DM or healthy controls (Figure 3B,C). As biological ageing affects the distribution of CD8^+^ T cells [15,16], we tested whether differences between IIM subtypes might be explained by age. Percentage of TEMRA cells and age were included in a model of linear regression revealing no statistically meaningful association across the age span of our cohort and corresponding TEMRA numbers (Figure 3D). Given these notable alterations, we decided to further interrogate the TEMRA compartment in detail. Consequently, we investigated the specific phenotype of TEMRA cells analyzing CD226^+^, CD57^+^ and CD28^+^ expression (Figure 3E). In IBM, TEMRA cells positive for CD226^+^ were decreased while cells positive for CD57^+^ were increased, both without reaching statistical significance (Figure 3F). However, TEMRA cells that lost CD28 (CD28^−^) were markedly expanded in IBM as compared to ASyS, DM and healthy.

To further dissect the phenotype of TEMRAs in IIM, we assessed markers of immune cell exhaustion including PD-1, TIM3, and LAG3. Surprisingly, exhaustion markers were unchanged on TEMRA cells from IBM patients, while DM patients displayed reduced levels of PD-1^+^ TEMRAs when comparing relative cell numbers (Figure 4A). Besides T cell exhaustion, we were interested in the migratory capacity of TEMRA cells in IIM as immune cell infiltration is a pathogenic hallmark in IIMs. CD18 constitutes the lymphocyte function-associated antigen 1 (LFA-1) together with CD11 and mediates T cell activation and adhesion [17]. CD18^+^ TEMRA cells were markedly expanded in IBM and CD18 expression (measured by mean fluorescence intensity) was increased (Figure 4A–C). Next, cytotoxicity and immune stimulatory capacities were measured by staining cells for a large panel of cytokines including GM-CSF, IL-17A, IL-22, IL-4, IFNγ and TNFα (Figure 4D,E). Here, fewer GM-CSF^+^ TEMRA cells were detected for DM. In contrast, IFNγ and TNFα expressing TEMRAs were substantially expanded in IBM (Figure 4D,E). Together, our data corroborate the presence of highly differentiated, cytotoxic CD8^+^ TEMRA cells in IBM and underpin their ability for cellular adhesion and infiltration as evidenced by CD18 expression.

### 3.4. CD4 T Cell Composition Is Shifted in ASyS

Next, we aimed to understand changes to the CD4 T cell compartment in IIM. We investigated changes to naïve (CD3^+^, CD4^+^, CD27^+^, CD45RO^−^), CM (CD3^+^, CD4^+^, CD27^+^, CD45RO^+^) and EM CD4^+^ T cells (CD3^+^, CD4^+^, CD27^−^, CD45RO^+^) (Figure 5A). We performed quantitative comparisons of cell populations and observed a shift of CD4 T cells from naïve to EM in ASyS, while other IIM displayed no meaningful alterations of CD4 T cell composition (Figure 5B). To understand functional alterations of expanded EM CD4^+^ T cells, we investigated a large panel of cytokines (Figure 5C). Here, EM CD4^+^ T cells positive for IL-4 were decreased in ASyS, whereas IFNγ^+^ EM CD4^+^ T cells were expanded in IBM. Differentiation of naïve CD4^+^ T cells into T helper subtypes requires a sufficient cytokine environment [18]. As such, IL-4 primarily drives polarization towards the Th2 phenotype [19]. Given diminished levels of IL-4^+^ CD4^+^ EM cells in ASyS, we hypothesized that the associated Th1/Th2 differentiation might be altered. Th1, Th2, Th17 and Tfh cells were analysed by a combination of CCR4, CCR6, CXCR3, ICOS and PD-1 surface expression (Appendix A). Indeed, in-depth analysis of helper T cell subsets revealed marked alterations in ASyS with Th1/Th2 balance skewed towards the Th1 phenotype (Appendix A). Th17 cells were comparable among the IIM spectrum. Intriguingly, Tfh cells were expanded in ASyS underlining aberrance of helper T cells in this condition. Taken together, our data argue that the framework of T helper cells is skewed in IIM.

### 3.5. Expansion of Distinct B Cell Subsets in IIM

Besides the T cell compartment, we aimed to understand B cell alterations in IIM. We defined distinct cell populations including naïve (CD19^+^, CD20^+^, IgD^+^, CD27^−^), not-switched memory (CD19^+^, CD20^+^, IgD^+^, CD27^+^), switched memory (CD19^+^, CD20^+^, IgD^−^, CD27^+^), CD27−/IgD− B cells (CD19^+^, CD20^+^, IgD^−^, CD27^−^) and regulatory B cells (Breg, CD19^+^, CD20^+^, IgD^−^, CD27^−^, CD38^+^, CD24^+^) (Figure 6A). Here, naïve B cells were more abundant in ASyS, while switched memory B cells were decreased compared to DM (Figure 6B). Other B cell populations remained unchanged. Biparametric downstream analysis might be insufficient to understand changes to the complex B cell continuum in IIM, particularly as a standardized classification based on flow cytometric markers is currently lacking for the B cell compartment. We employed the unsupervised clustering algorithm FlowSOM [11] for high-dimensional analysis of the B cell compartment. This approach aims to distinguish distinct B cell populations based on the identification of multiparametric markers. Indeed, application of FlowSOM clustering (Figure 6C) using expression profiles of 9 B cell markers identified a total of 12 corresponding meta-clusters (Figure 6D). This approach delineated a substantial expansion of FlowSOM meta-cluster (FSOM) 4 in DM, while FSOM 8 was decreased (Figure 6E). The expanded FSOM 4 cluster corresponds to IgG4^high^, IgG1^low^, CD38^high^ not-switched memory-like B cells, whereas FSOM 8 cluster mainly reflects a IgG1^low^, CD80^low^ naïve-like B cell population with varying IgG4 expression. Thus, B cell pathology is driven by distinct alterations in IIM with ASyS displaying a shift towards a naïve phenotype while B cell pathology in DM is characterized by IgG4/CD38 abundance.

### 3.6. Specific Immune Signatures Differentiate the Spectrum of IIM

Finally, we asked whether immune signatures identified in our analysis distinguish ASyS, DM and IBM patients. To this end, we selected key characteristics defining CD8, CD4 and B cell pathology across the IIM spectrum. We included TEMRA, CD28^−^ TEMRA, CD18^+^ TEMRA, IFNγ^+^ TEMRA and TNFα^+^ TEMRA numbers for the CD8 compartment; naïve CD4, EM CD4, Th1, Th2 and Tfh cells for the CD4/T helper compartment; and naïve B cells, IgG4^high^, IgG1^low^, CD38^high^ not-switched memory-like and IgG1^low^, CD80^low^ naïve-like B cells for the B cell compartment. These datasets were entered in a multivariable analysis using a principal component analysis (PCA) approach (Figure 7A). Each patient was treated as one data point. Indeed, specific immune features were able to distinguish IIM disorders. This resolution was attributed to specific features of immune pathology for IIM (Figure 7B): A terminally differentiated CD8 T cell pathology defined IBM, while ASyS was characterized by an altered CD4/T helper cell compartment. B cell pathology was evident for both ASyS and DM, but not IBM. Taken together, our analysis delineates specific changes to the peripheral immune compartment capable of separating diseases of the IIM spectrum.

## 4. Discussion

Disease outcomes and treatment responses vary substantially across the spectrum of IIM [1,20]. Thus, inflammatory pathways and mechanisms of disease chronicity likely diverge between IIM subtypes. However, treatment approaches remain mostly unspecific with glucocorticoids as first line choice [6]. Potentially owing to the rarity of the disease and the difficulty of routinely obtaining target tissue, i.e., standardized muscle biopsies, our understanding of immune signatures across the IIM spectrum remains incomplete. To develop specific treatment strategies according to clinical and immunological subtypes, further insight into the immune topography of IIM is needed.

First, ASyS is characterized by antibodies against the aminoacyl-transferase RNA synthetases, commonly against Jo-1, OJ, PL-7 or PL-12. To better understand disease specific pathways of autoimmunity, we investigated ASyS pathology in a previous study [21]. Here, we identified an extra-medullary inflammatory micro-milieu maintaining activated B cells. In line with our previous observations, in this study peripheral B cells were shifted towards a naïve/immature phenotype when compared to other IIM subtypes. Further, the CD4 lineage was shifted towards EM CD4^+^ T cells with suppressed expression of IL-4 compared to DM and IBM. In the context of the Th1/Th2 dichotomy, IL-4 mediates polarization towards Th2 [19,22]. Indeed, potentially owing to insufficient IL-4 stimulation, cells were shifted to the Th1 phenotype in ASyS. Further investigation of T helper cells revealed a pronounced expansion of Tfh cells. These cells are primarily found in secondary lymphoid organs [23]. The B cell follicle homing receptor CXCR5 enables Tfh to localize with B cells. Functionally, Tfh cells are critically involved in mediating B cell survival, activation and differentiation into plasma cells within germinal centres [9,24]. Two previous studies reported elevated levels of circulating Tfh cells in IIM (PM and DM patients) compared to healthy controls [25,26]. Further, we previously substantiated the presence of Tfh cells in ASyS muscle in comparison to non-diseased controls [21]. Promoting antigen-specific B cells [27], Tfh cells might provide a pro-stimulatory framework for B cell pathology in ASyS. Collectively, our findings argue that studying B cells is likely to contribute to our understanding of ASyS pathophysiology.

In DM, clinical presentation is highly variable. Characteristic skin lesions often precede proximal muscle weakness [28]. However, muscle affection is sometimes missing, a condition called amyopathic DM [28]. Moreover, DM might affect a number of target tissues, such as the skin, lung, joints or more rarely the heart. Several antibodies have been detected in association with DM, most notably anti-Mi2^+^ (5–20% of patients), anti-TIF1y^+^ (15–25%), anti-MDA5^+^ (7–15%) and anti-NXP2^+^ (15–25%) [20,28]. An IFN type 1 response is thought to drive inflammation in DM as evidenced in blood [29] and muscle [30]. While serological findings have been widely replicated, previous phenotyping studies of peripheral immune cells were inconsistent [31,32,33]. Differences between studies might be attributed to clinical distinctions, including treatment status (naïve vs. treated), age at onset (adult vs. juvenile) or clinical presentation (myopathic vs. amyopathic). As such, treatment naïve patients displayed reduced CD4 and CD8 T cell counts that were ameliorated upon corticosteroid treatment [31]. In line, broad immune cell subsets remained unchanged in our analysis, potentially owing to treatment status. Moreover, a flow cytometric analysis of 13 DM patients reported a shift towards Th2 cells in DM as compared to PM [32]. We corroborate this finding when comparing Th2 cells from DM and ASyS patients. Intriguingly, using an unsupervised approach, we also observed a pronounced expansion of an IgG4^high^, IgG1^low^, CD38^high^ not-switched memory-like cell cluster in DM, while a naïve B cell-like population was decreased. CD38 is highly expressed on plasma cells as well as germinal centre B cells and implicated in B cell activation and survival [34]. A previous study, albeit in juvenile DM, reported expansion of a CD24^high^, CD38^high^ transitional B cell population corresponding to a type 1 IFN signature [35]. Thus, CD38 might be important for sustaining B cell pathology in DM. In respect to IgG4, Th2 cells secreting IL-4 and IL-13 mediate isotype switching to IgG4 on B cells [36,37]. We argue that expanded Th2 cells [32] might provide a framework for IgG4 switched B cells in DM. Functional consequences of IgG4 in DM remain enigmatic as IgG4 have been associated with both anti-inflammatory effects but also pathology as evidenced in IgG4-related disease [38,39]. Further research into the Th2/IgG4-axis in the context of DM is warranted to better understand potential implications for disease chronicity.

Increasing evidence argues for T cell mediated autoimmunity as primary mechanism underlying IBM, which has shaped recent debate and anticipates CD8 T cell depletion as potential tool for disease amelioration (discussed in detail by Greenberg [4]). Indeed, cytotoxic CD8^+^ T cells have been identified in IBM as early as 1988 [40] and since then been confirmed in blood and muscle [41,42]. Specifically, these cells are clonally expanded, terminally differentiated and cytotoxic [43,44,45]. Moreover, these cells have been characterized by the loss of CD28 and gain of CD57 and KLRG1 besides markers of cytotoxicity such as IFNγ [43,44,45]. A recent immunophenotyping study corroborated expanded, KLRG1^+^ TEMRA cells in blood of IBM patients, while KLRG1^+^ NK cells remained unchanged [46]. We underline these findings and report a marked expansion of the TEMRA subset differentiating IBM from other entities of IIM. While we also observed loss of CD28, upregulation of CD57 did not reach statistical significance in comparison with ASyS and DM patients. Interestingly, TEMRAs in IBM do not show alterations of markers of T cell exhaustion (PD-1, TIM3, LAG3 [47,48,49]). While the scope of our study is not focused on T cell-mediated cytotoxicity, previous data suggest that T cell differentiation status correlates to cytotoxicity [44]. Conversely, the ultrastructural target of T cell cytotoxicity remains speculative. However, as mononuclear infiltrates of myofibers consist mostly of CD8 T cells in IBM [44], it might be reasonable to speculate that CD8 T cell cytotoxicity is directed against skeletal muscle cells. Moreover, we also observed that TEMRAs in IBM strongly express CD18. Together with CD11, CD18 forms the integrin LFA-1 mediating T cell activation and migration [17,50,51]. ICAM-1 engages LFA-1 and enables leukocyte extravasation [51]. We speculate that the upregulation of the LFA-1 axis reflects the potential for tissue infiltration of CD8^+^ T cells in IBM. Albeit limited to two IBM patients, application of natalizumab inhibiting VCAM-1/VLA-4 interaction and leukocyte migration abolished endomysial inflammation in a pilot study [52]. Natalizumab did, however, not result in clinical improvement as measured by the MMT-8 [52]. Recently, the treatment of IBM patients with rapamycin in a phase 2 study did not reach its primary endpoint as change to the maximal voluntary isometric knee extension [53]. However, efficacy in secondary outcome measures supported further investigation of rapamycin in a phase 3 trial [53,54]. Interestingly, rapamycin inhibits VCAM-1 signalling and leukocyte infiltration in vivo [55]. Thus, it is tempting to pursue leukocyte infiltration as a potentially understudied therapeutic target in IBM.

As limitation, the scope of this study is restricted to the analysis of immune cells from peripheral blood. Immune cell patterns in the muscle compartment might be distinct. Further, a potential limitation of this study might be introduced by epidemiological and treatment differences between individual patients and patient groups. To reduce bias, only patients with confirmed diagnosis according to EULAR/ACR Classification were included and therapies were required to be stable. However, it should be noted that a potential treatment effect is difficult to exclude, particularly as a high proportion of IBM patients were treatment naïve as compared to ASyS and DM. In respect to antibody status, this study does not provide statistical power to account for potential differences between antibody subtypes. A strength of this study is its size and depth providing insights into the main lymphocyte populations concurrently in an informative cohort, particularly as large-scale studies comparing IIM subtypes are sparse.

Thus, our study provides a comprehensive overview of the peripheral immune topography of the IIM spectrum and new insights into potential drivers of disease. Application of a high-dimensional flow cytometry approach appears to be a valuable tool to detect overt and even subtle differences among pathologies. These differences are sufficiently distinct to differentiate the peripheral immune response across the studied entities. In essence, we argue that immune cell pathologies distinguish the IIM spectrum and might serve as target for more specific treatment approaches. ASyS appears to be characterized by CD4/T helper perturbations supporting B cell-mediated autoimmunity, while DM displays distinct alterations of the B cell compartment. A pathology of terminally differentiated CD8^+^ T cells distinguishes IBM from other IIM.

## Figures and Tables

**Figure 1 cells-11-03330-f001:**
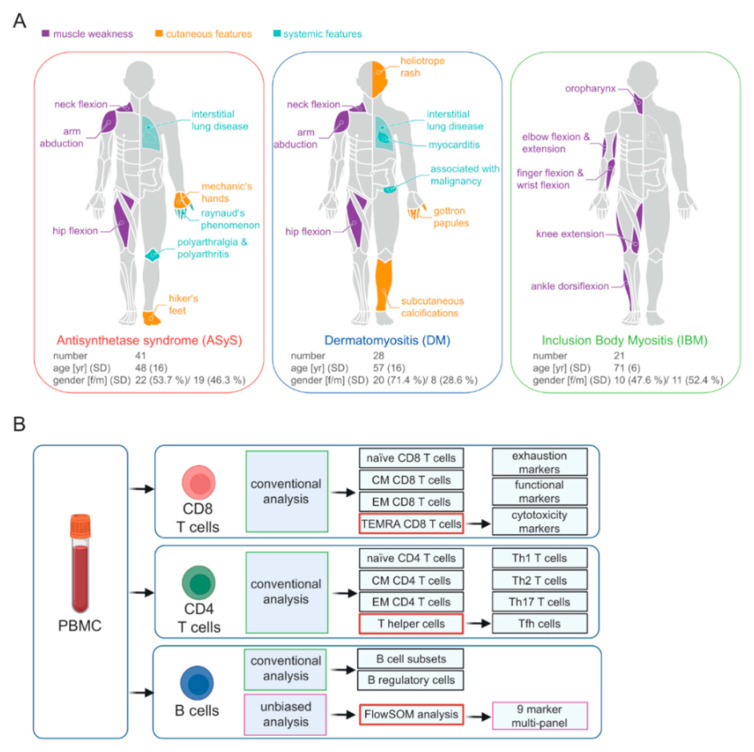
Overview of the patient cohort and analyzed flow cytometry panels. (**A**) Clinical phenotypes and characteristics of ASyS, DM and IBM. (**B**) Overview of flow cytometry panels analysed in the study. PBMC were analysed by flow cytometry. The CD8 and CD4 T cell compartment were analysed by conventional gating. TEMRA CD8 T cells and T helper cells were then analysed in further detail. B cells were investigated by conventional analysis and by unbiased analysis using FlowSOM. Abbreviations: ASyS = anti-synthetase syndrome; DM = dermatomyositis; IBM = inclusion body myositis.

**Figure 2 cells-11-03330-f002:**
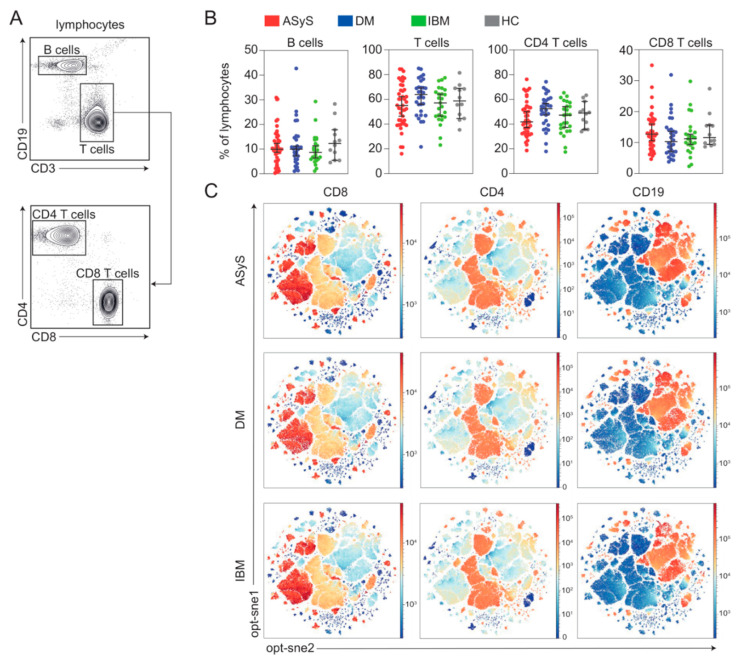
Broad immune cell patterns are unchanged in IIM. (**A**) Representative flow cytometry gating strategy for immune cell subsets. (**B**) Frequencies of broad immune cell subsets in IIM and in healthy controls. (**C**) Global optimization of tSNE (opt-sne) projection of CD8, CD4 and B cells (CD19) for all patients concatenated and overlaid. Expression of indicated proteins is colour coded. Each point represents an individual ASyS (red), DM (blue), IBM (green) patient or healthy controls (grey). Abbreviations: ASyS = anti-synthetase syndrome; DM = dermatomyositis; IBM = inclusion body myositis; tSNE = t-distributed stochastic neighbour embedding.

**Figure 3 cells-11-03330-f003:**
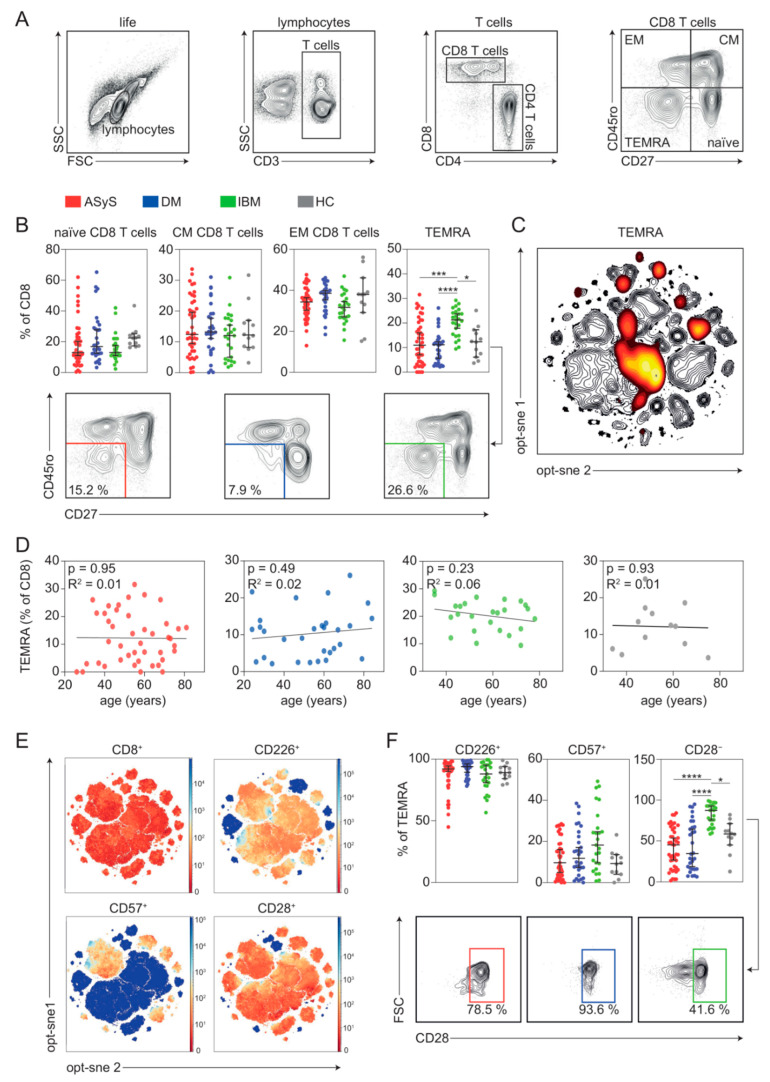
Topography of the CD8 T cell population in IIM. (**A**) Representative flow cytometry gating strategy for immune cell subsets. (**B**) Frequencies of CD8 T cell subsets in IIM and representative flow cytometry plots. (**C**) Global optimization of tSNE (opt-sne) projection of the CD8 T cell population for all patients concatenated and overlaid. The TEMRA subset is overlaid in red. (**D**) Linear regression model displaying association of TEMRA levels as % of CD8^+^ T cells and age. (**E**) Global opt-sne projection of CD8^+^ T cells for all patients concatenated and overlaid. Expression of indicated proteins is colour coded. (**F**) Frequencies of indicated TEMRA cells and representative flow cytometry plots. Each point represents an individual ASyS (red), DM (blue), IBM (green) patient or healthy controls (grey). Significance was determined by Kruskal–Wallis test: * *p* < 0.05, *** *p* < 0.001, and **** *p* < 0.0001. Abbreviations: ASyS = anti-synthetase syndrome; CM = central memory; DM = dermatomyositis; EM = effector memory; IBM = inclusion body myositis; TEMRA = terminally differentiated effector memory T cells; tSNE = t-distributed stochastic neighbour embedding.

**Figure 4 cells-11-03330-f004:**
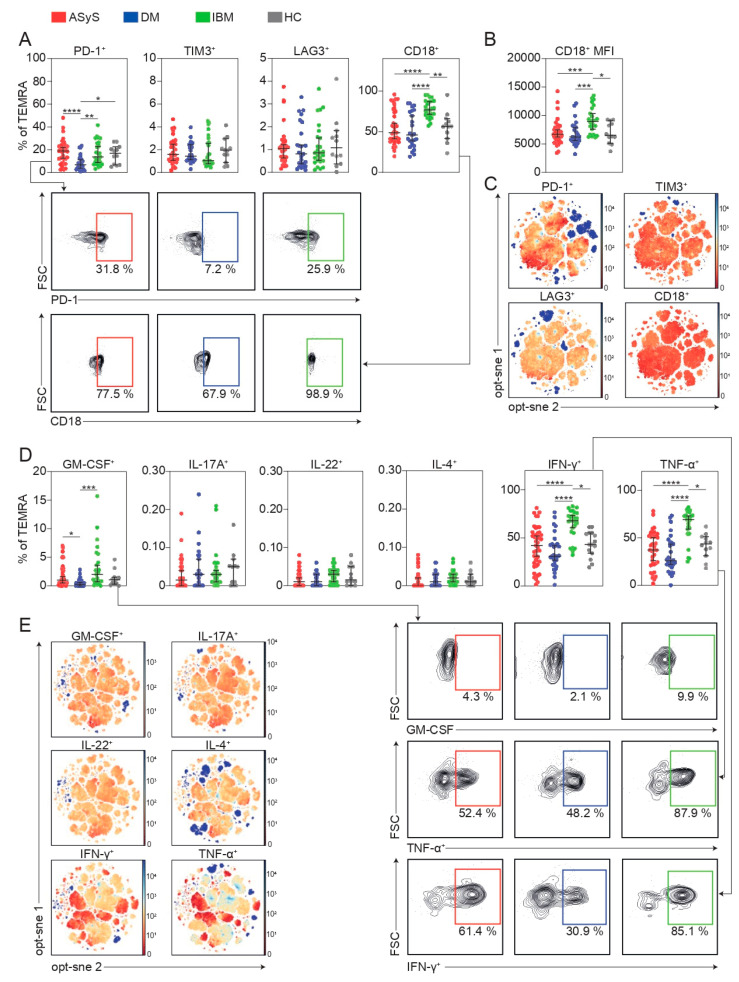
IBM is characterized by distinct changes to the TEMRA population. (**A**) Frequencies of indicated TEMRA cells and representative flow cytometry plots. (**B**) MFI of surface CD18 on TEMRA cells. (**C**) Global optimization of tSNE (opt-sne) projection of the CD8 T cell population for all patients concatenated and overlaid. Expression of indicated proteins is colour coded. (**D**) Frequencies of indicated TEMRA cells and representative flow cytometry plots. Each point represents an individual ASyS (red), DM (blue), IBM (green) patient or healthy controls (grey). (**E**) opt-sne projection of the CD8 T cell population for all patients concatenated and overlaid. Expression of indicated proteins is colour coded. Significance was determined by Kruskal–Wallis test: * *p* < 0.05, ** *p* < 0.01, *** *p* < 0.001, and **** *p* < 0.0001. Abbreviations: ASyS = anti-synthetase syndrome; DM = dermatomyositis; IBM = inclusion body myositis; MFI = mean fluorescence intensity; TEMRA = terminally differentiated effector memory T cells; tSNE = t-distributed stochastic neighbour embedding.

**Figure 5 cells-11-03330-f005:**
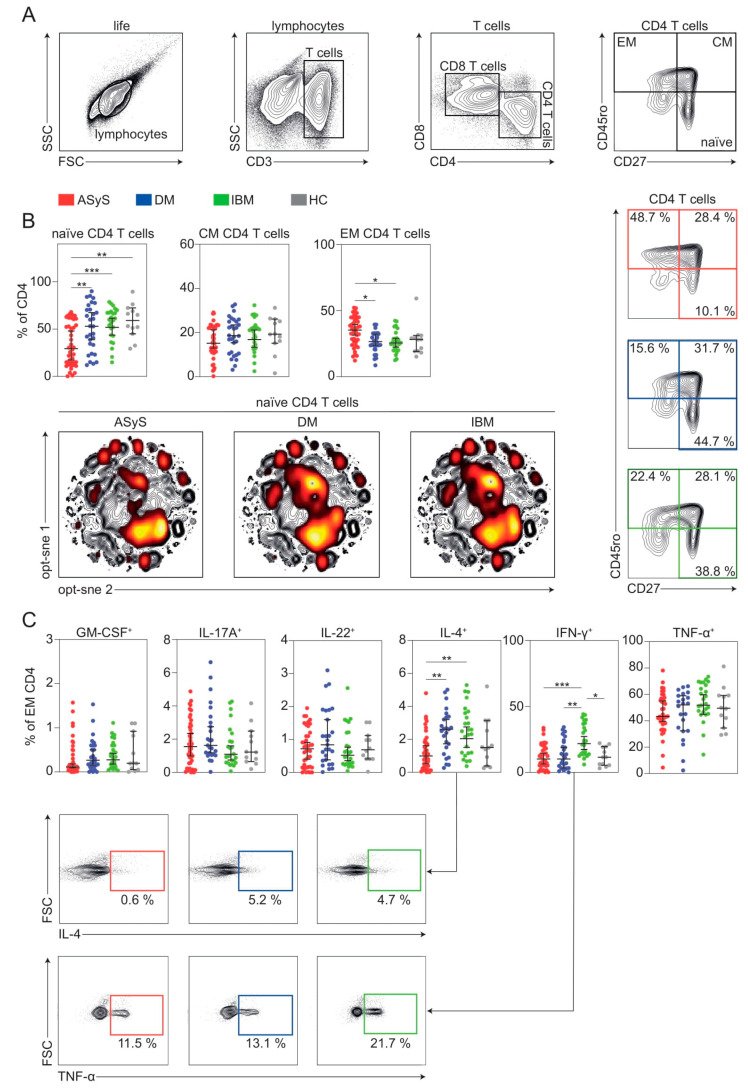
CD4 T cell subsets are distorted in IIM. (**A**) Representative flow cytometry gating strategy for immune cell subsets. (**B**) Frequencies of indicated TEMRA cells and representative flow cytometry (top). Global optimization of tSNE (opt-sne) projection of the CD4 T cell population for all patients concatenated and overlaid. Naïve CD4 T cells are indicated in red in each condition (bottom). (**C**) Representative flow cytometry gating strategy for immune cell subsets. Each point represents an individual ASyS (red), DM (blue), IBM (green) patient or healthy controls (grey). Significance was determined by Kruskal–Wallis test: * *p* < 0.05, ** *p* < 0.01, *** *p* < 0.001. Abbreviations: ASyS = anti-synthetase syndrome; CM = central memory; DM = dermatomyositis; EM = effector memory; IBM = inclusion body myositis.

**Figure 6 cells-11-03330-f006:**
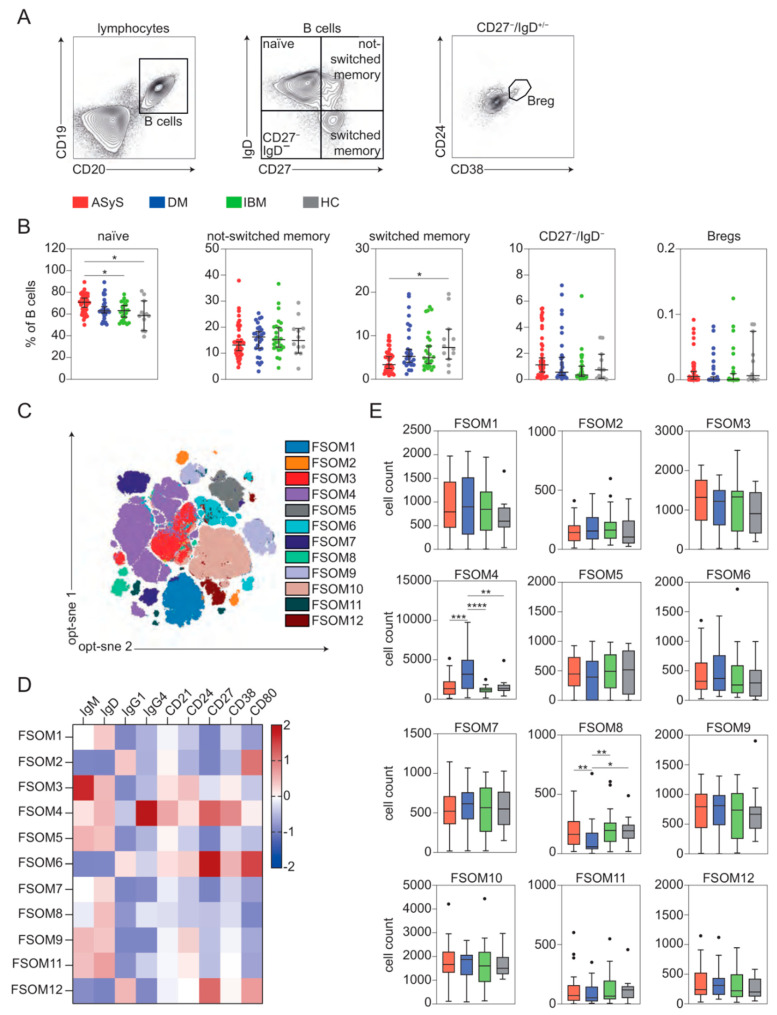
B cell dysregulation is a feature of IIM. (**A**) Representative flow cytometry gating strategy for immune cell subsets. (**B**) Frequencies of indicated B cell subsets. (**C**) Global optimization of tSNE (opt-sne) projection of the B cell population with meta-clusters identified by FlowSOM clustering superimposed. First, flow cytometric markers displayed in (**D**) were entered into opt-sne for generation of a two-dimensional graph of the dataset. Next, the FlowSOM algorithm was applied to the same dataset for unbiased identification of distinct cell clusters. A total of 115 clusters were acquired. Consensus hierarchical clustering identified 12 distinct FlowSOM meta-clusters (FSOM) across these clusters. Each cell visualized in the opt-sne map was labelled to the corresponding FSOM. (**D**) Heatmap displaying mean MFI for indicated FSOM clusters and corresponding marker (column-scaled z-scores). (**E**) Box plots displaying cell numbers of indicated FSOM clusters. Each point represents an individual ASyS (red), DM (blue), IBM (green) patient or healthy controls (grey). Significance was determined by Kruskal–Wallis test: * *p* < 0.05, ** *p* < 0.01, *** *p* < 0.001, and **** *p* < 0.0001. Abbreviations: ASyS = anti-synthetase syndrome; CM = central memory; DM = dermatomyositis; EM = effector memory; IBM = inclusion body myositis.

**Figure 7 cells-11-03330-f007:**
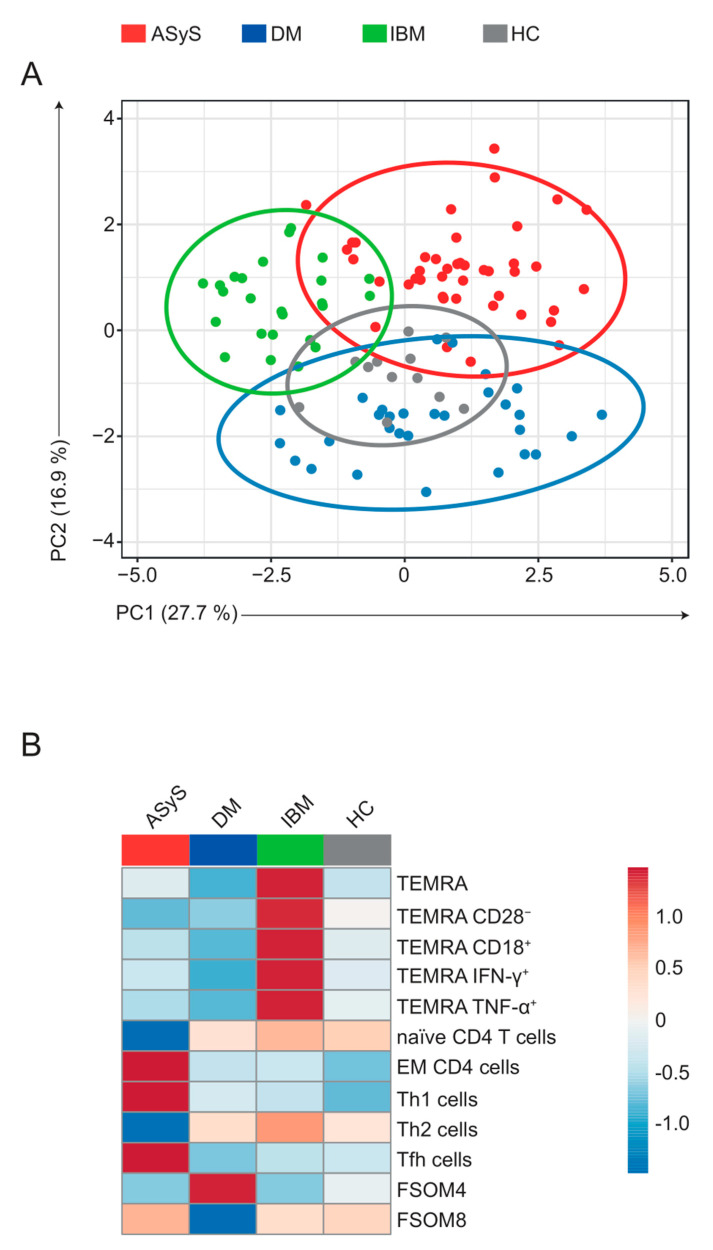
Multivariable analysis distinguishes IIM subtypes. (**A**) PCA of key features of ASyS, DM, IBM and healthy controls. SVD was used to calculate principal components. X and Y axis show principal component 1 and principal component 2, respectively. Prediction ellipses are such that with probability 0.8, a new observation from the same group will fall inside the ellipse. Data were centred and scaled. The PCA was constructed using the features displayed in (**B**). (**B**) Heatmap displaying mean values for indicated features (column-scaled z-scores). Each point represents an individual ASyS (red), DM (blue), IBM (green) patient or healthy controls (grey). Abbreviations: ASyS = anti-synthetase syndrome; DM = dermatomyositis; EM = effector memory; IBM = inclusion body myositis, PCA = principal component analysis; SVD = singular value decomposition.

**Table 1 cells-11-03330-t001:** Demographics and baseline disease characteristics. Disease duration was defined as the timespan between symptom onset and the date of blood acquisition. For group comparisons a χ^2^-test for categorial or a Kruskal–Wallis analysis of variance with post hoc Bonferroni-testing for continuous data were conducted; ns *p* ≥ 0.05, * *p* < 0.05, ** *p* < 0.01, *** *p* < 0.001. Abbreviations: ASyS = anti-synthetase syndrome, AZA = azathioprine, CK = creatine kinase, CYP = cyclophosphamide, DM = dermatomyositis, GCS = glucocorticosteroids, IBM = inclusion body myositis, IVIG = intravenous immunoglobulins, MMF = mycophenolate mofetil, MMT-8 = manual muscle testing of 8 muscle groups, MTX = methotrexate, N = number, ns = non-significant.

Diagnosis	ASyS (*n* = 41)	DM (*n* = 28)	IBM (*n* = 21)	Healthy Control	Group Differences
Age, years, median (range)	48 (38–81)	57 (42–79)	71 (52–85)	49 (22–79)	IBM vs. DM **, IBM vs. ASyS ***, IBM vs. Healthy control ***, DM vs. ASyS *
Female, N (%)	22 (53.7)	20 (71.4)	10 (47.6)	6 (50)	ns
Disease duration, years (SD)	3.3 (4.3)	3.5 (6.2)	6.1 (5.1)	None	ns
MMT-8, median	147	140	127	150	IBM vs. ASyS ***, IBM vs. healthy cont.rols ***
CK level (U/l)	3055 (4241)	1207 (2829)	491 (374)	119 (198)	ASyS vs. healthy control ***, DM vs. healthy control *
Malignancies, N (%)	12 (29.3%)	14 (50%)	5 (23.8%)	None	ASyS vs. healthy control *, DM vs. healthy control **
Treatment, N (%)					
None	13 (31.7)	7 (25)	9 (42.3)	12 (100)	
GCS	11 (26.8)	4 (14.3)	2 (9.5)	-	
AZA	1 (2.4)	3 (10.7)	-	-	
MMF	-	-	-	-	
IVIG	1 (2.4)	1 (3.6)	12 (57.1)	-	
MTX	6 (14.6)	4 (14.3)	-	-	
CYP	3 (7.3)	1 (3.6)	-	-	
AZA + GCS	2 (4.9)	6 (21.4)	-	-	
IVIG + GCS	-	2 (7.1)	-	-	

## Data Availability

All relevant data is presented in this manuscript. Raw data will be provided upon reasonable request from qualified researchers.

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
