# Peer review of "High-Dimensional Cytometry Dissects Immunological Fingerprints of Idiopathic Inflammatory Myopathies"

_cells, 2022, doi:10.3390/cells11203330_

Round 1

Reviewer 1 Report

This interesting, well presented, paper illustrates the application of high dimensional flow cytometry to 3 forms of inflammatory myopathies. It reports some interesting differences that could be useful for developing appropriate therapies.

Comments:

1.    Is it necessary to abbreviate antibody to ‘ab’?  Similarly ‘HC’ for healthy control. Personally I think not, having them in full would make it easier to read. There are several abbreviations used already.

2.    The inclusion criteria for the controls would be better in the methods, rather than the results

3.    The alignment in the ‘treatment’ section of table 1 needs a little adjustment

4.    Limitations of the study are mentioned but is ‘stable treatment’ sufficient to exclude a possible effect, especially as several IBM patients were not receiving treatment

5.    In addition, there is no correlation with the antibody in the DM patients. Is this possible. Also the presence/absence of malignancies.

6.    How many of the patients had muscle biopsies? Is  a correlation with inflammation and cell types in the biopsies possible to do?

7.    Do the authors plan to study juvenile dermatomyositis?

8.    Do the authors consider that their method could be used for aiding diagnosis as well as designing therapies

Author Response

Please see the attachment for the full PBP.

Reviewer 2 Report

This manuscript reports on unsupervised immunophenoytypic analyses of lymphocyte subsets in different idiopathic inflammatory myopathies. It demonstrates several anomalies that may help distinguish between the different entities explored.

Globally, the manuscript is highly redundant between the introduction and discussion and between the text and tables. A more synthetic and easier to follow formulation should be used. Conversely, although the original use of recently-developed AI methodology for unsupervised flow analyses has been wisely used, the uneducated reader will be unable to understand what has actually been performed. A flow chart of the different investigations and subtypes assessed would be very helpful. A complete, detailed description of how a t-sne graph must be interpreted is lacking. The rationale for moving between global unsupervised analyses and manual gating of biparametric scattergrams (some of them disputable) should be provided. The use of FlowSOM is utterly opaque.

The attempt at providing absolute numbers using beads with highly processed cells (Ficoll Hypaque, deep-freezing, thawing, washes, permeabilization…) is totally irrelevant.

Based on these complex pieces of information, it is very difficult to derive a clear message from what however appears to be a large amount of potentially interesting data.

Minor

The paragraph on cell staining is largely incomplete, does not mention how cytokine production was assessed and does not explain the immunophenotypes investigated. The complete list of antibodies used provided in supplemental material is not very helpful as it does not provide panels nor the relevance of each marker.

The proportion of patients without immunotherapy is in fact relatively low and the sentence should be modified page 3. Table 1 is awkwardly presented and highly redundant with information provided jut above in the text. This should be harmonized to better convey messages in the text and details in the table.

Figure 1A should be individualized

Please define TEMRA at first occurrence

Author Response

Please see the attachment for our PBP.

Round 2

Reviewer 2 Report

The manuscript by C. Nelke et al. has been greatly improved by complying to the reviewers comments. Especially the new figure 1 is extremely useful. A few little things remain to be amended, after thorough reading of this new version, as detailed below.

Abstract

-         Compartments not compartment

-         Immunoglobulin G4 (IgG4)

M&M

-          Ninety not 90 as sentence beginning, idem twelve later on

-         Spell out EULAR ACR and ENMC

-         Indicate the city and country (or state for the USA) of all manufacturers at first mention, then only manufacturer name

-         Replace 2D by bidimensional

-         Opt-sne graphs (plural)

Results

-         Use consistently CD45RO and CD45RA (not CD45ro or CD45ra)

-         Change “Among the CD8 T cell populations, naïve, CM and EM CD8 T cell populations remained unchanged…” by “Naïve, CM and EM CD8 T cell populations remained unchanged,…”

-         Do not repeat the definition of opt-sne after the first occurrence except in figure legends

-         It would be nice to see a scattergram with CD45RA expression since so much emphasis is brought on TEMRA cells

Discussion

-         CD20 is downregulated on plasma-cells. Although the use of anti-CD20 could be supported by the anomalies of B-cells evidenced in this study, the discussion around plasma-cells should be minimized.

-         IgG4 have been (in the sentence about functionality of these cells

-         The cytotoxic potential of CD8+ cells has not been demonstrated here. A more prudent wording should be used to discuss this point, end perhaps also evoke towards which cells could this cytotoxicity be exerted.

-         Expand on this enigmatic sentence :” Clinical readout parameter remained un-changed”

Author Response

We appreciate the valuable feedback. Please see our attached PBP below.
